# Genomic Association Analysis of Growth and Backfat Traits in Large White Pigs

**DOI:** 10.3390/genes14061258

**Published:** 2023-06-13

**Authors:** Peng Zhou, Chang Yin, Yuwei Wang, Zongjun Yin, Yang Liu

**Affiliations:** 1Department of Animal Genetics and Breeding, College of Animal Science and Technology, Nanjing Agricultural University, Nanjing 210095, China; 2022105032@stu.njau.edu.cn (P.Z.); 2021105019@stu.njau.edu.cn (C.Y.); 2022805132@stu.njau.edu.cn (Y.W.); 2College of Animal Science and Technology, Anhui Agricultural University, Hefei 230036, China; yinzongjun@ahau.edu.cn

**Keywords:** backfat thickness, growth, GWAS, pig, meta-analysis

## Abstract

The pig industry is significantly influenced by complex traits such as growth rate and fat deposition, which have substantial implications for economic returns. Over the years, remarkable genetic advancements have been achieved through intense artificial selection to enhance these traits in pigs. In this study, we aimed to investigate the genetic factors that contribute to growth efficiency and lean meat percentages in Large White pigs. Specifically, we focused on analyzing two key traits: age at 100 kg live weight (AGE100) and backfat thickness at 100 kg (BF100), in three distinct Large White pig populations—500 Canadian, 295 Danish, and 1500 American Large White pigs. By employing population genomic techniques, we observed significant population stratification among these pig populations. Utilizing imputed whole-genome sequencing data, we conducted single population genome-wide association studies (GWAS) as well as a combined meta-analysis across the three populations to identify genetic markers associated with the aforementioned traits. Our analyses highlighted several candidate genes, such as *CNTN1*—which has been linked to weight loss in mice and is potentially influential for AGE100—and *MC4R*, which is associated with obesity and appetite and may impact both traits. Additionally, we identified other genes—namely, *PDZRN4*, *LIPM*, and *ANKRD22*—which play a partial role in fat growth. Our findings provide valuable insights into the genetic basis of these important traits in Large White pigs, which may inform breeding strategies for improved production efficiency and meat quality.

## 1. Introduction

The Large White pig breed holds a prominent place in pig production due to its favorable traits, such as its efficient feed conversion, rapid growth, high slaughter value, and having a significant proportion of lean meat. These characteristics have made it a preferred choice for breeding and meat production. Selective breeding over time has led to significant improvements in genetic traits—particularly in terms of rapid growth and enhanced leanness. Consequently, the Large White pig breed has gained a significant market share in the pig breeding industry in recent decades [1]. Days of age at 100 kg live weight (AGE100) is a moderately heritable trait, with a heritability ranging from 0.3 to 0.5, and serves as a genetic indicator for evaluating growth rate [2]. Backfat thickness at 100 kg (BF100) is a key predictor for assessing lean meat rate in pigs [3], as there exists a strong negative genetic correlation between backfat thickness and lean meat rate [4]. The traits of AGE100 and BF100 hold significant economic significance in Large White pigs; therefore, acquiring a deeper understanding of the genetic architecture underlying growth rate and fat deposition traits can facilitate advancements in pig breeding and genetic enhancement.

Genome-wide association analysis (GWAS) is a comprehensive approach that explores genetic variation—particularly single nucleotide polymorphisms (SNPs)—across the entire genome to identify variants potentially associated with specific traits. In livestock breeding, GWAS serves as a valuable tool for identifying economically important traits such as growth, carcass [5,6,7], reproduction [8,9], immunity [10,11] and flesh quality [12,13], among other candidate genes studied. While GWAS is commonly employed to identify genetic markers associated with complex traits in livestock, it has some limitations, such as low reproducibility [14] and a high incidence of false positives [15]. To overcome these limitations, meta-analysis is often employed, as it enables the detection of subtle effects and the analysis of larger sample sizes. Furthermore, meta-analysis is well-suited for the investigation of rare variants in sequencing studies and can be particularly advantageous when dealing with heterogeneity in combined studies [16].

This study employed a GWAS approach to investigate the genetic factors underlying AGE100 and BF100 in three distinct populations of Large White pigs, each characterized by unique genetic backgrounds. The GWAS analysis allowed for the identification of significant SNP loci associated with these traits. Subsequently, a comprehensive meta-analysis was conducted, integrating data from all three populations, to further validate and identify noteworthy SNPs and candidate genes associated with AGE100 and BF100. Through our analysis, we uncovered the potential involvement of *CNTN1* as a candidate gene that could influence AGE100, drawing from its association with weight loss in mice. Furthermore, the gene *MC4R*—which has been linked to obesity and appetite—emerged as a possible determinant affecting both traits. Additionally, our findings implicated *PDZRN4*, *LIPM*, and *ANKRD22* as partial contributors to fat growth. The outcomes of this study provide valuable insights into the intricate genetic underpinnings of these complex traits—thereby enhancing our understanding of and paving the way for advancements in pig breeding and overall improvement strategies.

## 2. Materials and Methods

### 2.1. Ethics Statement

The Animal Welfare Committee of Nanjing Agricultural University conducted a review of all the animal testing and sample collection techniques used in this research. This review process included a careful examination of the ethical considerations of the research, as well as the methods and procedures used to ensure the safety and welfare of the animals involved. The Committee approved the animal testing and sample collection techniques used in this research, ensuring that the animals were treated humanely, and that the data collected was accurate and reliable (Permit number: DK652).

### 2.2. Feeding of Experimental Animals

In total, there were 2295 individuals of three different pig breeds including 500 Canadian, 295 Danish, and 1500 American Large White pigs. The Large White pigs used in this research were all in the fattening stage and were fed in large columns, with a feeding density of about 0.8–1.2 m^2^ per pig depending on weight. Wet curtain fans and heat preservation lamps were used to control the building temperature at 19–21 °C; the relative humidity was kept at 40–70% and ventilation was 0.3 m/s; harmful gases such as NH_3_ were kept at 20 mg/m^3^ or less, H_2_S at 8 mg/m^3^ or less, and CO at 5 mg/m^3^ or less. Lighting conditions were 100 Ix and the lighting time was 8 h/d. Feeding was carried out by automatic troughs to ensure that there would be no break in feeding. Rodents were controlled by rodenticide to ensure that they did not come into contact with the pigs, and the pigs were enclosed by gauze nets to ensure that wild birds did not enter the barn. All experimental animal buildings were disinfected with the same procedure, and all experimental animals were vaccinated with the same swine fever vaccine.

### 2.3. Animals and Phenotypes

Pigs from all the pig populations were selected for analysis, taking individuals of approximately 160 days of age with similar feeding conditions, both healthy and disease-free. These pigs were managed under standardized feeding conditions and measured individually at an average weight of 100 kg. To evaluate backfat thickness, ultrasound was utilized to measure between the 11th and 12th ribs at the same weighing time. Using the equation corrected for the measurement AGE100 [17]: AGE100 = $measured\;age+\frac{100\mathrm{kg}-measured\;age}{CF}$, where CF_male_ = $\frac{measured\;weight}{measured\;age}\;*\;1.826$ and CF_female_ = $\frac{measured\;weight}{measured\;age}\;*\;1.715$, and also using the equation corrected for BF100 [18]: BF100_male_ = $measured\;back\;fat\;*\;\frac{12.402}{12.402+0.106\;*\;\left(measured\;weight-100\right)}$ and BF100_female_ = $measured\;back\;fat\;*\;\frac{13.705}{13.705+0.119\;*\;\left(measured\;weight-100\right)}$.

### 2.4. Genotyping and Quality Control

This study utilized three distinct populations of Large White pigs—including 500 Canadian, 295 Danish, and 1500 American pigs—as experimental materials. All individuals with phenotypes were genotyped using the GeneSeek GGP porcine HD array. According to *Sus scrofa* version 10.2, the SNP chip consisted of 50,915 probes, and autosomes were further upgraded to the latest version of the porcine genome—*Sus scrofa* version 11.1. The final remaining autosomal SNPs were 34,150 Kb for the Canadian line, 34,543 Kb for the Danish line, and 34,497 Kb for the American line.

Exclusion of individuals that did not meet Hardy-Weinberg equilibrium was carried out. Quality control was performed by PLINK (V1.90; http://www.cog-genomics.org/; accessed on 16 March 2023) [19]. Pigs with call rates < 0.9 were excluded, and SNPs with minor allele frequencies (MAF) below 0.05 and call rates < 0.9 were excluded in each population species.

### 2.5. Population Genome Analysis

Eigenvalues and eigenvectors were obtained using PLINK (v1.90; http://www.cog-genomics.org/; accessed on 16 March 2023), and principal component analysis (PCA) was performed using the remaining SNPs [19]. Additionally, linkage disequilibrium (LD, denoted as r^2^) was calculated in each line. In this study, *SNeP* software (v1.1; https://bioinformaticshome.com/tools/descriptions/SNeP.html#gsc.tab=0; accessed on 16 March 2023) was used to calculate the population effect size (*Ne*) [20].

### 2.6. Genome-Wide Analysis

The linear mixed model used for GWAS was as follows:y=μ+Xb+Wg+Zu+e
where y is the vector of the target phenotypes of individuals; μ is the overall mean; b is the vector of fixed effects: sex (two levels) and year–season, in which seasons were comprised of four levels (Spring: March to May; Summer: June to August; Autumn: September to November; Winter: December to February); g is the vector of the SNP effects, X is the matrix of incidence associating each observation to the pertinent level of fixed effects, W is the incidence matrix relating observations to SNP effects with elements coded as 0, one, and two for genotype A_1_A_1_, A_1_A_2_, and A_2_A_2_, respectively; u is the random additive genetic effect of the individual and is assumed to be distributed as $N\left(0,\;G\sigma_{u}^{2}\right),$ G is the genomic relationship matrix and $\sigma_{u}^{2}$ is the polygenic additive genetic variance, Z  is the incidence matrix for u, and e is the random residual, assumed to be distributed as $N\left(0,\;I\sigma_{e}^{2}\right)—$where I  is the identity matrix and $\sigma_{e}^{2}$ is the residual variance. Associations between the target traits and the SNPs were analyzed using single-SNP association tests in each population, which were implemented by the mlma option of the software GCTA (Version 1.93.3 beta; http://yanglab.westlake.edu.cn; accessed on 16 March 2023) [21].

In addition, a meta-analysis was implemented through METAL (version 2011; https://csg.sph.umich.edu/abecasis/Metal/download/; accessed on 16 March 2023) [22]. In the meta-analysis, the weighted Z-score model took account of the *p*-values, the direction of the SNPs’ effects, and the number of individuals. In each case, threshold *p*-values were set to −log_10_ (1/SNPs) and −log_10_ (0.05/SNPs) for suggestive and Bonferroni-adjusted genome-wide significance, respectively. The Bonferroni threshold for significant SNPs were determined as 1.449 × 10^−6^, 1.465 × 10^−6^, and 1.448 × 10^−6^ for the Canadian, American, and Danish line, respectively. A quantile-quantile (QQ) plot of −log_10_ (*p*-values) was examined to determine how well GCTA accounted for population structure and family relatedness.

### 2.7. Candidate Gene Annotation and Functional Enrichment Analysis

To identify potential candidate genes associated with the significant SNP loci, we employed the BioMart database (http://www.ensembl.org/; accessed on 16 March 2023) for comprehensive annotation. In our study, candidate genes were selected within a 500 Kb genomic region surrounding the significant SNPs, taking into account both upstream and downstream regions. Additionally, we utilized the R package WebGestaltR [23] to perform functional annotation of genes located in these regions of interest. The functional annotation involved examining Gene Ontology (GO) terms and Kyoto Encyclopedia of Genes and Genomes (KEGG) pathways associated with the candidate genes. To facilitate this analysis, we employed the Metascape database (https://metascape.org/; accessed on 16 March 2023), which provides a valuable resource for exploring the functional characteristics of genes. Furthermore, to supplement our investigations, an extensive literature search was conducted to gather relevant information regarding the functions and roles of these candidate genes. This holistic approach allowed us to thoroughly explore and evaluate the potential functions and pathways associated with the identified candidate genes, providing a solid foundation for further exploratory studies in this field.

## 3. Results

### 3.1. Phenotypic Data and Population Structure Analysis

Table 1 presents the average values of AGE100 and BF100 for the three strain groups. The data revealed that the Canadian line had a higher average AGE100 than the Danish and American line groups, whereas the Canadian line group had a relatively lower BF100. These results suggest that the Canadian line group exhibited greater leanness but comparatively lower growth efficiency, as compared to the other two strain groups.

A principal component analysis was conducted to assess the genetic variation indices of the three populations. The results demonstrated significant differences in the genetic backgrounds of the American, Canadian, and Danish line pigs, with PC1 effectively separating the three populations (Figure 1A). Additionally, Figure 1B depicts the average LD (r^2^) at varying physical distances between two motifs on all autosomes of the three populations. The data indicated a strong linkage disequilibrium in all three populations at an SNP physical distance of approximately 0.5 Mb, averaged over the autosomes. Furthermore, Figure 1C illustrates that the Danish and Canadian lines had similar effective population sizes until generation 100.

### 3.2. SNP Loci Associated with AGE100

The study identified two suggestive significant SNPs at three loci in the American lineage population, located on *Sus scrofa* chromosomes 2 and 14 (SSC2, SSC14). Additionally, two potentially significant SNPs were detected in the Canadian lineage population, located on SSC1 and SSC6. Additionally, there was one significant SNP on SSC13. Furthermore, one SNP locus located on SSC5 (SSC5: 52888587) appeared at significant levels in the Danish line. Through the meta-analysis, a total of 10 potentially significant SNPs were identified, with seven located in SSC1 and the remaining three located in SSC2, SSC5, and SSC14, respectively (Figure 2, Table 2).

### 3.3. SNPs Significantly Associated with BF100

The study detected six SNPs loci with potentially significant levels of BF100 in the American population, with one located on SSC5 and five on SSC16. Similarly, two potentially significant SNPs were identified in the Canadian population, with one located on SSC1 and the other on SSC5. In contrast, only one potentially significant SNP was detected on SSC1 in the Danish population. Through a meta-GWAS analysis of the three populations, the study found that only two SNPs on SSC1 were significantly associated with BF100, while the remaining nine SNPs were detected at potentially significant levels. Of these, six were located on SSC1, two were located on SSC5, and one was located on SSC18 (Figure 3, Table 3).

### 3.4. GO and KEGG Analysis

In the present study, the Gene Ontology (GO) analysis revealed a significant enrichment of molecular functions associated with BF100 in the meta-analysis (Figure 4A). The candidate genes exhibited significant enrichment in molecular functions related to protein-containing complex binding, as well as enrichment in growth factor activity, RNA binding, and other functions. Additionally, a Kyoto Encyclopedia of Genes and Genomes (KEGG) enrichment analysis was conducted to gain further insights into the pathways associated with these genes (Figure 4B).

For AGE100, the meta-analysis revealed a significant enrichment of biological processes based on Gene Ontology (GO) analysis (Figure 4C). The candidate genes exhibited significant enrichment in molecular functions related to the positive regulation of neurogenesis, positive regulation of nervous system development, and positive regulation of cell development. Additionally, KEGG enrichment analysis was conducted to gain further insights into the pathways associated with these genes (Figure 4D).

## 4. Discussion

The present study aimed to assess the population structure of Large White pigs originating from distinct genetic backgrounds—namely, the Canadian, American, and Danish lines. Through an extensive analysis, we observed consistent patterns of linkage disequilibrium (LD) among the three populations. However, notable population stratification was identified in all three groups based on the analysis of effective population size (*Ne*). The Danish and Canadian lines demonstrated similarity only within the initial 100 generations; these findings were further corroborated by the principal component analysis, which clearly highlighted the presence of population stratification across the three populations. Thus, our comprehensive investigation provides compelling evidence of distinct population structures within the Large White pig breeds under study.

In the context of GWAS, the identification of significant SNP loci within a single population is a crucial step, while subsequent meta-analysis serves as a means to validate and refine these loci. Particularly in studies involving large sample sizes and complex traits with subtle effects, this approach becomes invaluable. However, our study’s findings suggest that significant SNP loci identified within a single population may not always be consistently detected in a meta-analysis that combines multiple populations. This discrepancy can be attributed to variations in LD patterns across different populations—thereby leading to the identification of distinct sets of significant SNPs. Consequently, the pooling of populations for meta-analysis can result in attenuated associations between SNPs and the studied traits. These observations underscore the importance of carefully considering population-specific LD patterns when conducting meta-analyses and interpreting the collective results in order to gain a comprehensive understanding of the genetic architecture underlying complex traits.

In the analysis of AGE100, we identified a highly significant SNP locus (SSC6_154765159) in the Canadian population, specifically located on chromosome 6. Further investigation through Biomart annotation of this SNP locus revealed the presence of a *DAB1* gene positioned approximately 164,281 base pairs downstream from the SNP locus. Notably, previous studies in mice have highlighted the involvement of the *DAB1* gene in neurodevelopmental processes [24]. *DAB1* acts as an adaptor protein crucial for the intracellular transduction of Reelin signaling—thereby regulating the migration and differentiation of post-mitotic neurons during brain development. These findings suggest that *DAB1* may play a pivotal role in the genetic mechanisms underlying AGE100, emphasizing its potential significance in shaping growth and developmental processes in Large White pigs [25,26].

In our analysis of the Danish lines, we identified a SNP (SSC 5_72888587) that exhibited potential significance in both the GWAS and meta-analysis. The location of this SNP on the genome revealed the presence of a *CNTN1* gene located 720,076 base pairs upstream from the SNP. The *CNTN1* gene encodes a cell adhesion molecule known as contactin-1, which belongs to the immunoglobulin-superfamily. Studies in mice have associated *CNTN1* with weight loss [27] and anorexia [28]. Functionally, *CNTN1* is a glycosylphosphatidylinositol (GPI)-anchored neuronal membrane protein that acts as a ligand for Notch, thereby promoting oligodendrocyte maturation and myelin formation. Deficiencies in *CNTN1* can lead to impairments in inhibitory synaptic development [29]. Furthermore, a previous study on tumor cells suggested that *CNTN1* affects the cell cycle and cell growth, potentially influencing the trait under investigation [30]. The SNP locus of interest was found within the *PDZRN4* gene, which has been identified as a highly expressed gene in human abdominal fat and a potential candidate gene associated with intramuscular fat content in pigs [31]. These findings suggest that both *CNTN1* and *PDZRN4* may contribute to the regulation of AGE100 in Large White pigs, although further research is needed to elucidate the exact mechanisms underlying their effects.

Furthermore, the Biomart analysis of SNPs detected through the meta-analysis revealed the presence of the *LIPM* gene located at 463,324 base pairs upstream of a potentially significant SNP (SSC14_101189804). The *LIPM* gene belongs to the lipase I.3 subfamily [32]. Additionally, we observed the annotation of a partially overlapping gene—*ANKRD22*. *ANKRD22* is a novel human N-myristoylated protein, and its expression product is specifically localized in lipid droplets [33]. Both the *LIPM* and *ANKRD22* genes are likely to have an impact on lipid growth—a factor that may influence AGE100 in pigs. These findings suggest the potential involvement of *LIPM* and *ANKRD22* in the regulation of AGE100 through their roles in lipid metabolism.

In our meta-analysis, a noteworthy SNP locus (SSC1_160773437) was identified that corresponds to the melanocortin 4 receptor (*MC4R*) gene, based on Biomart gene annotation. The *MC4R* gene, known as a retinoid-like gene, exhibits expression in multiple tissues and has been implicated in the regulation of obesity-related metabolic diseases [34]. Moreover, this gene is recognized for its crucial involvement in energy homeostasis and the control of appetite [35]. Given its known functions, the *MC4R* gene is likely to exert an influence on weight gain and AGE100 in pigs by modulating feed intake. The identification of the *MC4R* gene as a candidate associated with AGE100 suggests its potential as a key regulatory element affecting growth characteristics in pigs.

The potentially significant SNP (SSC1_160773437) identified in the meta-analysis assay was annotated as being within the *MC4R* gene, which has also been associated with obesity. This finding suggests that the *MC4R* gene may play a role in regulating fat content growth and dorsal label thickness—specifically in relation to the BF100 trait. Furthermore, a gene encoding the melanocortin-2 receptor accessory protein (*MRAP2*) was identified at a location 205,846 bp upstream of the significant SNP (SSC1_53461852) in the Canadian population. This gene has been reported to have an important role in regulating appetite and maintaining energy homeostasis. *MC4R* can interact with melanocortin receptor accessory protein 2 (*MRAP2*) and form a modulation complex [36]. Additionally, genetic variations in *MRAP2* have been associated with obesity [37,38]. This observation suggests that the *MRAP2* gene may contribute to the regulation of BF100 in Large White pigs. What is more, we annotated some of the SNP loci detected in the single population analysis and meta-analysis to genes related to cell proliferation and cell cycle—for example, *ZC3HAV1* [39], *CCND2* [40], and *MAP3K1* [1]. *ZC3HAV1* is described as a zinc finger antiviral protein, playing vital roles in immunity and microRNA-mediated stress responses [41]. The *CCND2* gene is a strong candidate gene for backfat deposition in pigs, supported by changes in expression and its involvement in adipogenesis [42]. Additionally, *CCND2* could have an influence through growth-related processes on conformation traits in Danish pigs [43]. *MAP3K1* is an important regulator of T cell proliferative expansion and cell cycle progression [44]. This gene may have an impact on adipocyte growth regulation as well as proliferation, affecting backfat thickness.

## 5. Conclusions

In this study, our objective was to investigate the genetic factors underlying AGE100 and BF100 in purebred Large White pigs from Canadian, American, and Danish-line breeds. We conducted single-population GWAS and mixed-population meta-analysis to identify SNPs associated with these traits. Our analyses revealed significant associations between SNPs and candidate genes, including the *MC4R* gene, which showed potential significance and potential effects on appetite and obesity regulation in both AGE100 and BF100 traits. We also identified candidate genes such as *PDZRN4*, *LIPM*, and *ANKRD22* that were associated with fat growth. Additionally, we detected the *CNTN1* gene, which exhibited significant associations in the Danish line population. However, further investigations are warranted to fully understand the role of *CNTN1* in body weight regulation in Large White pigs. These findings contribute to our understanding of the genetic architecture of these traits and provide valuable insights for future research focused on enhancing pig breeding and genetic improvement strategies.

## Figures and Tables

**Figure 1 genes-14-01258-f001:**
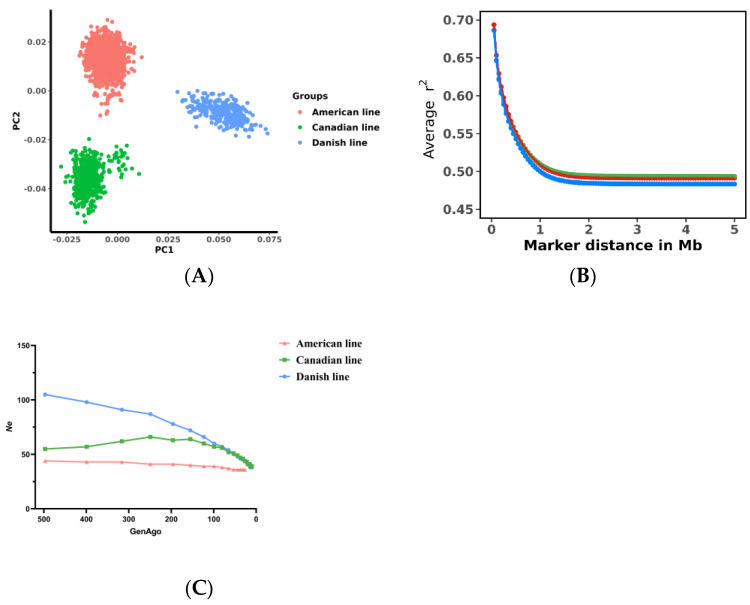
Descriptive statistics of population structure. (**A**) Principal component analysis showing the relationship between the two principal components (PC1, PC2) and the proportion of genetic variance explained (percentage of variation explained) in pigs of the American, Canadian, and Danish lines. (**B**) Linkage disequilibrium (LD) in pigs of American, Canadian, and Danish lines, with r^2^ values averaged over an interval of 0.5 Mb between physical distances of paired SNPs and combined on autosomes. (**C**) Mean estimated effective population size (Ne) based on numbers plotted over the last 500 generations.

**Figure 2 genes-14-01258-f002:**
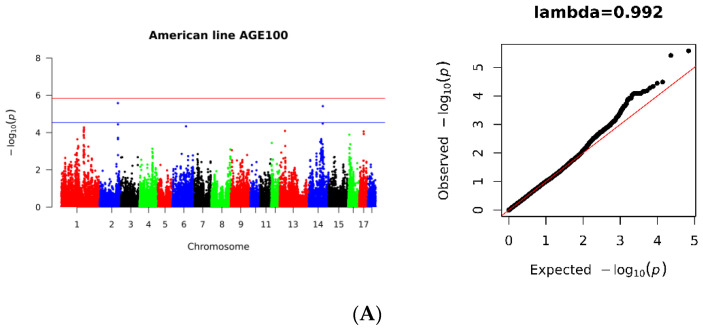
Manhattan and Quantile–Quantile (QQ) plots for genome-wide association analysis of the AGE100 trait. (**A**) American line, (**B**) Danish line, (**C**) Canadian line. (**D**) Three population meta-GWAS analysis. *x*-axis indicates autosomes and *y*-axis indicates −log_10_ (*p*-value).

**Figure 3 genes-14-01258-f003:**
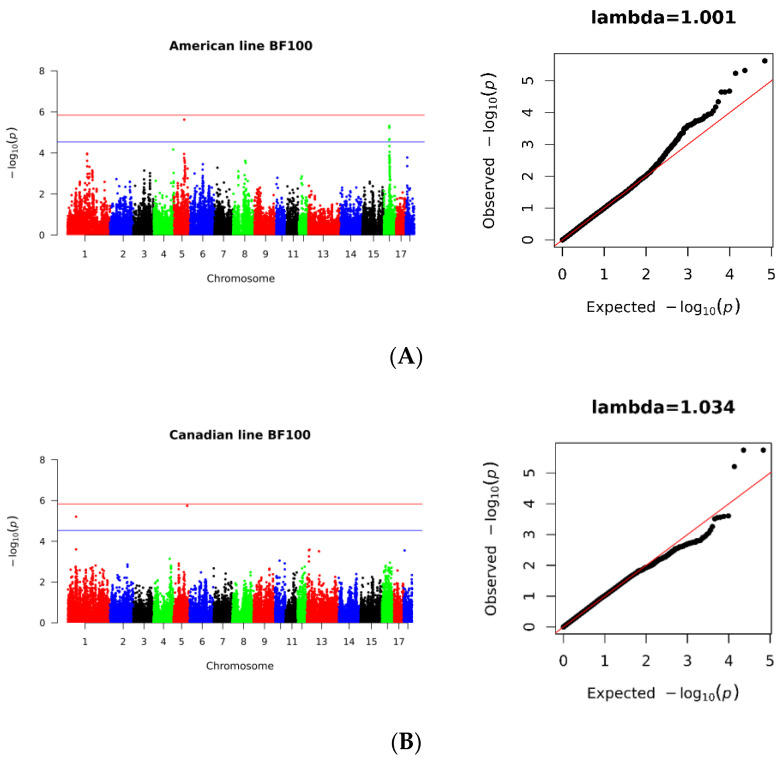
Manhattan and Quantile–Quantile (QQ) plots for genome-wide association analysis of the BF100 trait. (**A**) American line, (**B**) Danish line, (**C**) Canadian line. (**D**) Three population meta-GWAS analysis. *x*-axis indicates autosomes and *y*-axis indicates −log_10_ (*p*-value).

**Figure 4 genes-14-01258-f004:**
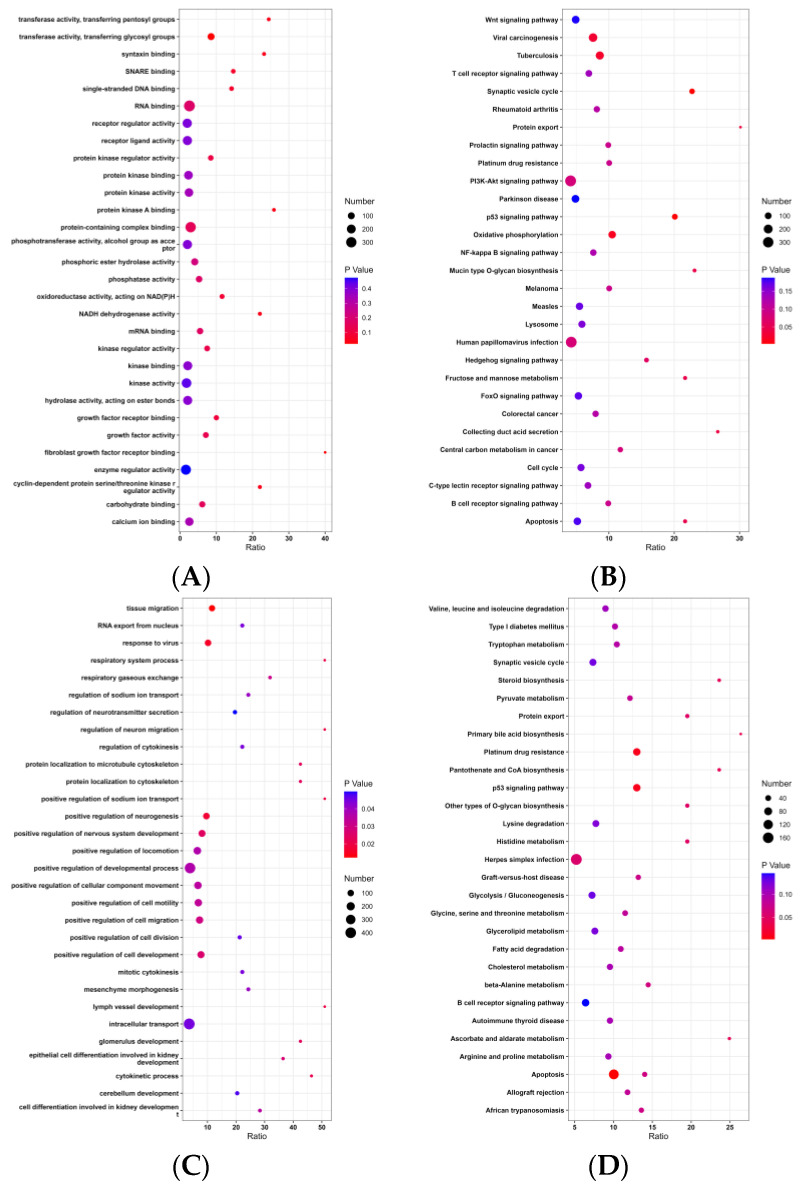
GO and KEGG analysis of BF100 and AGE100 traits in the meta-analysis. (**A**) GO analysis of BF100. (**B**) KEGG analysis of BF100. (**C**) GO analysis of AGE100. (**D**) KEGG analysis of AGE100.

**Table 1 genes-14-01258-t001:** Phenotypic data for three Large White pig populations.

Source	N	Day_Mean ± Std	bf_Mean ± Std	SNP_Num	After Filling SNP
Canadian line	500	180.78 ± 17.57	12.14 ± 2.47	34,150	3,837,195
Danish line	314	167.29 ± 15.16	18.84 ± 4.4	34,543	12,437,523
American line	1500	161.48 ± 13.21	13.72 ± 2.77	34,497	8,077,147

**Table 2 genes-14-01258-t002:** Significant SNPs and genes in which they are located, identified in the genome-wide association study for the AGE100 trait.

SSC (*Sus scrofa* Chromosome)	Position (bp)	*p*-Value	Distance	Genes
American line				
14	101,189,804	3.83 × 10^−6^	Upstream 463,324 bp	*LIPM*
14	101,189,804	3.83 × 10^−6^	Upstream 441,847 bp	*ANKRD22*
Danish line				
5	72,888,587	5.92 × 10^−6^	Upstream 720,076 bp	*CNTN1*
5	72,888,587	5.92 × 10^−6^	Within	*PDZRN4*
Canadian line				
6	154,765,159	5.96 × 10^−7^	Downstream 664,281 bp	*DAB1*
Meta-analysis				
1	160,773,437	1.38 × 10^−5^	Within	*MC4R*
5	72,888,587	5.92 × 10^−6^	Upstream 720,076 bp	*CNTN1*
5	72,888,587	5.92 × 10^−6^	Within	*PDZRN4*
14	101,189,804	4.13 × 10^−6^	Upstream 463,324 bp	*LIPM*
14	101,189,804	4.13 × 10^−6^	Upstream 441,847 bp	*ANKRD22*

**Table 3 genes-14-01258-t003:** Significant SNPs and genes in which they are located, identified in the genome-wide association study for the BF100 trait.

SSC (*Sus scrofa* Chormosome)	Position (bp)	*p*-Value	Distance	Gene
American line				
16	35,401,235	4.77 × 10^−6^	Downstream 470,895bp	*MAP3K1*
Canadian line				
1	53,461,852	6.19 × 10^−6^	Upstream 205,846bp	*MRAP2*
Meta-analysis				
1	160,773,437	3.98 × 10^−6^	Within	*MC4R*
1	161,632,403	2.86 × 10^−6^	Within	*CPLX4*
18	10,662,063	3.29 × 10^−6^	Upstream 103,166bp	*ZC3HAV1*

## Data Availability

Not applicable.

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
