# Peer review of "Genomic Association Analysis of Growth and Backfat Traits in Large White Pigs"

_genes, 2023, doi:10.3390/genes14061258_

Round 1

Reviewer 1 Report

Manuscript Genes-2420016 entitled “Genome-wide association analysis for growth and backfat traits in Large White pigs by imputed data. Please notice the following:

General view: The manuscript revealed a good research point of view that was expressed in good English and grammar. The manuscript must be subjected to a certain degree of simplification to catch the normal reader's mind with some modifications and little rephrasing and copyediting to enhance the readability and understanding of the text. 

Title: Clear to a greater extent but preferred to be modified into “Genomic Association Analysis of Growth and Backfat Traits in Large White Pigs”.

Abstract: Clear to a greater extent. A few modifications and a little rephrasing are suggested to enhance the readability and understanding of the text.

Keywords: Rearrange in alphabetical order.

Introduction: Clear, informative, comprehensible, and properly displayed. A few modifications and a little rephrasing and copyediting are suggested to enhance the readability and understanding of the text.

The aim: Clear and informative.

Materials and Methods: Please notice the following:

1.      Provide the institutional approval number and statement.

2.      Illustrate the microclimatic conditions of the experimental animals such as housing system and design, temperature, relative humidity, ventilation system, lighting system, feeding & watering systems, drainage system, rodent control measures, pest control measures, wild bird control measures, cleaning and disinfection procedures, and treatment and vaccination act if any during the study.

3.      A few modifications and a little rephrasing and copyediting are suggested to enhance the readability and understanding of the text.

Results: Novel, clear, and informative with a necessity to carry out a certain degree of simplification, rephrasing, and copyediting to enhance the readability and understanding of the text.

Discussion: Concise, clear, comprehensible, informative, and contribute to knowledge with a moderate level of speculation and comparison. A few modifications and a little rephrasing and copyediting are suggested to enhance the readability and understanding of the text.

Conclusion: Clear and informative.

Authors’ contributions: Clear and informative.

Funding: Clear and informative.

Acknowledgment: Clear and informative.

References: Sufficient as 51.4% (18 out of 35) were published in the past five years.

Tables: Well organized and presented.

Figures: Well organized and presented.

The manuscript was expressed in good English and grammar. Minor Editing is requested to simplify the text up to a certain level to catch the reader's mind.

Author Response

  1. Title: Clear to a greater extent but preferred to be modified into “Genomic Association Analysis of Growth and Backfat Traits in Large White Pigs”.

Response:

Thanks for recommending a better title. Modified as suggested.

  1. Abstract: Clear to a greater extent. A few modifications and a little rephrasing are suggested to enhance the readability and understanding of the text.

Response:

Thanks for precious suggestion. Modified as suggested (Line 9 – line 25).

  1. Keywords: Rearrange in alphabetical order.

Response:

Thanks for precious suggestion. Modified as suggested.

  1. Introduction: Clear, informative, comprehensible, and properly displayed. A few modifications and a little rephrasing and copyediting are suggested to enhance the readability and understanding of the text.

Response:

Thanks for precious suggestion. Modified as suggested (Line 29 – line 69).

  1. Materials and Methods: 

1) Provide the institutional approval number and statement.

2) Illustrate the microclimatic conditions of the experimental animals such as housing system and design, temperature, relative humidity, ventilation system, lighting system, feeding & watering systems, drainage system, rodent control measures, pest control measures, wild bird control measures, cleaning and disinfection procedures, and treatment and vaccination act if any during the study.

3) A few modifications and a little rephrasing and copyediting are suggested to enhance the readability and understanding of the text.

Response:

Thanks for precious suggestion. 1) Added as suggested (Line72 – line 78). 2) Added as suggested (Line80 – line 92). 3) Modified as suggested.

  1. Results: Novel, clear, and informative with a necessity to carry out a certain degree of simplification, rephrasing, and copyediting to enhance the readability and understanding of the text.

Response:

Thanks for precious suggestion. Modified as suggested (Line 167 – line 252).

  1. Discussion: Concise, clear, comprehensible, informative, and contribute to knowledge with a moderate level of speculation and comparison. A few modifications and a little rephrasing and copyediting are suggested to enhance the readability and understanding of the text.

Response:

Thanks for precious suggestion. Modified as suggested (Line 255 – line 343).

Reviewer 2 Report

This is an interesting manuscript aimed to identify significant genomic SNPs and candidate genes associated with AGE100 and BF100 in Large White pig populations. However, there are some major issues to consider in order to improve the manuscript.

1. Introduction:

·      Please verify that the objective is the same as described in the abstract.

2. Materials and Methods:

-       2.1. Animals and phenotypes:

·      More data about pig populations should be interesting (i.e., age, body condition, etc.).

-       2.2. Genotyping and quality control:

·      How many individual were genotyped in total?, and how many from each breed?

·       “Sus scrofa” should be in Italic style (please correct through the manuscript).

·      What is the electronic address to access PLINK software?

·      Why the authors did not include Hardy-Weinberg equilibrium test as quality control?

-       2.3. Population genome analysis:

·      What is the electronic address to access SNeP software?

-       2.4. Genome-wide analysis:

·      Which were the assumptions of the genomic model?

·      What is the electronic address to access GCTA software?

·      What is the electronic address to access METAL software?

-       2.5. Candidate gene annotation:

·      Why the authors did not include Gene Ontology and Pathway Enrichment analyses?

·      The citation reference number should not be in Italic style.

3. Results:

·      Please explain why the authors are considering SNPs as significant even though they did not comply with Bonferroni test adjustment?

·      I suggest indicating the Bonferroni threshold for significant SNPs according to each pig breed.

4. Discussion:

·      I suggest explaining with more detail the biological implications of each identified gene on the phenotypic traits with which they were associated.

·      I guess more references should be useful to support findings of this study.

5. Conclusions:

·      This section appears to be a brief summary of the results.

·      I suggest including 3 or more conclusive sentences describing whether the results obtained were sufficient to meet the objective of the study, as well as possible implications and recommendations for future work.

Author Response

  1. Introduction: Please verify that the objective is the same as described in the abstract.

Response:

Thanks for your precious reminding. Verified as suggested, the objective is both the same in introduction and abstract.

  1. Materials and Methods: Animals and phenotypes: More data about pig populations should be interesting (i.e., age, body condition, etc.)

Response:

Thanks for your precious suggestion. Added as suggested (Line 92 – line 95).

  1. Materials and Methods: Genotyping and quality control: How many individuals were genotyped in total? and how many from each breed?

Response:

Thanks for your precious suggestion. Added as suggested (Line 92- line93).

  1. Materials and Methods: Genotyping and quality control: “Sus scrofa” should be in Italic style (please correct through the manuscript).

Response:

Thanks for your precious suggestion. Corrected as suggested.

  1. Materials and Methods: Genotyping and quality control: What is the electronic address to access PLINK software?

Response:

Thanks for your precious suggestion. Added as suggested.

  1. Materials and Methods: Genotyping and quality control: Why the authors did not include Hardy-Weinberg equilibrium test as quality control?

Response:

Thanks for mentioning this. We did Hardy-Weinberg equilibrium test and added as suggested (Line 111).

  1. Materials and Methods: Population genome analysis: What is the electronic address to access SNeP software?

Response:

Thanks for your precious suggestion. Added as suggested.

  1. Materials and Methods: Genome-wide analysis: Which were the assumptions of the genomic model?

Response:

Thanks for your precious suggestion. The genomic model is linear mixed model (LMM), which follows the polygenic effect hypothesis.

  1. Materials and Methods: Genome-wide analysis: What is the electronic address to access GCTAsoftware?

Response:

Thanks for your precious suggestion. Added as suggested.

  1. Materials and Methods: Genome-wide analysis: What is the electronic address to access METALsoftware?

Response:

Thanks for your precious suggestion. Added as suggested.

  1. Materials and Methods: Candidate gene annotation: Why the authors did not include Gene Ontology and Pathway Enrichment analyses?

Response:

Thanks for your precious suggestion. Added as suggested (Line 235 – line 247).

  1. Materials and Methods: Candidate gene annotation: The citation reference number should not be in Italic style.

Response:

Thanks for your precious suggestion. Corrected as suggested.

  1. Results: Please explain why the authors are considering SNPs as significant even though they did not comply with Bonferroni test adjustment. I suggest indicating the Bonferroni threshold for significant SNPs according to each pig breed.

Response:

Thanks for your precious suggestion. We have added Bonferroni threshold for each pig breed (Line 144 – line 145). During the revision, we have found some mistakes, and corrected the location of SNPs (AGE100 of American line and Canadian line; BF100 of Canadian line).

  1. Discussion: I suggest explaining with more detail the biological implications of each identified gene on the phenotypic traits with which they were associated. I guess more references should be useful to support findings of this study.

Response:

Thanks for your precious suggestion. Modified as suggested (Line 254 – line 342).

  1. Conclusions: This section appears to be a brief summary of the results. I suggest including 3 or more conclusive sentences describing whether the results obtained were sufficient to meet the objective of the study, as well as possible implications and recommendations for future work.

Response:

Thanks for your precious suggestion. Modified as suggested (Line 344 – line 356).

Round 2

Reviewer 2 Report

The manuscript has improved significantly. All sections are clear and well supported. I suggest to accept the manuscript.